



# Shipborne MAX-DOAS measurements for validation of TROPOMI NO$_2$ products

Ping Wang[1], Ankie Piters[1], Jos van Geffen[1], Olaf Tuinder[1], Piet Stammes[1], and Stefan Kinne[2]

[1]Royal Netherlands Meteorological Institute (KNMI), De Bilt, The Netherlands
[2]Max Planck Institute for Meteorology, Hamburg, Germany

**Correspondence:** Ping Wang (ping.wang@knmi.nl)

**Abstract.** Tropospheric NO$_2$ and stratospheric NO$_2$ vertical columns are important TROPOMI data products. In order to validate the TROPOMI NO$_2$ products, KNMI MAX-DOAS instruments have measured NO$_2$ on ship cruises over the Atlantic and the Pacific oceans. The MAX-DOAS instruments have participated in five cruises on-board R.V. Sonne (in 2017 and 2019) and R.V. Maria S. Merian (in 2018). The MAX-DOAS measurements were acquired in 7 months and spanned about 300° in longitude and 90° in latitude. During the cruises there were also aerosol measurements from Microtops sun-photometers. The MAX-DOAS measured stratospheric NO$_2$ columns between $1.5 \times 10^{15}$ and $3.5 \times 10^{15}$ molec cm$^{-2}$, and tropospheric NO$_2$ up to $0.6 \times 10^{15}$ molec cm$^{-2}$. The MAX-DOAS stratospheric NO$_2$ vertical columns have been compared with TROPOMI stratospheric NO$_2$ vertical columns and the stratospheric NO$_2$ vertical columns simulated by TM5-MP model. Good correlation is found between the MAX-DOAS and TROPOMI and TM5 stratospheric NO$_2$ vertical columns, with a correlation coefficient of 0.93 or larger. The TROPOMI and TM5 stratospheric NO$_2$ vertical columns are about $0.4 \times 10^{15}$ molec cm$^{-2}$ higher than the MAX-DOAS measurements. The TROPOMI tropospheric NO$_2$ has also good agreement with the MAX-DOAS measurements. The tropospheric NO$_2$ vertical columns is as low as $0.5 \times 10^{15}$ molec cm$^{-2}$ over remote oceans.

## 1 Introduction

Nitrogen dioxide (NO$_2$) and nitrogen oxide (NO) – usually referred to as nitrogen oxides (NO$_x$ = NO + NO$_2$) – are air pollutant trace gases in the troposphere. The tropospheric NO$_2$ is mostly produced at high temperatures in combustion processes but also in soil microbial process and lightning events. In the stratosphere, NO$_2$ is an ozone-depleting substance produced primarily from the oxidation of nitrous oxide (N$_2$O) (Crutzen, 1970; Johnston, 1971; Seinfeld and Pandis, 2006). NO$_x$ can also suppress ozone depletion by converting reactive chlorine and hydrogen compounds into unreactive reservoir species (Murphy et al., 1993).

Stratospheric NO$_2$ total columns have a strong diurnal cycle which is caused by the sunlight-driven balance between NO and NO$_2$, and is influenced by (bounded to) a total NO$_x$ amount. At night, NO$_x$ is in the form of NO$_2$, which is oxidized by





$O_3$ to produce $NO_3$, and $NO_3$ is converted to $N_2O_5$ in the presence of $NO_2$. Therefore, $N_2O_5$ is produced at night and $NO_2$ decreases during night.

At daytime, the $NO_2$ and NO are in a photochemical balance: the photolysis of $NO_2$ into NO and the oxidation of NO into $NO_2$ via ozone. The stratospheric $NO_2$ decreases at sunrise, because photo-dissociation brings $NO_2$ back in balance with NO. The daytime $NO_2$ concentrations increase gradually, which is caused by the slow increase in total $NO_x$. The slow increase of $NO_x$ during the daytime is due to the photo-dissociation of $N_2O_5$. In the lower stratosphere, additional reactions involving formation of $HNO_3$ and $ClONO_2$ also affect the total $NO_x$ available.

Tropospheric $NO_2$ concentrations have been derived from Ultraviolet/Visible backscatter satellite spectrometers such as Global Ozone Monitoring Experiment (GOME) (Burrows et al., 1999), SCanning Imaging Absorption spectroMeter for Atmospheric CHartographY (SCIAMACHY) (Bovensmann et al., 1999), Ozone Monitoring Instrument (OMI) (Levelt et al., 2006), GOME-2 (Munro et al., 2006) and Ozone Mapping and Profiler Suite (OMPS) (Yang et al., 2014). The TROPOspheric Monitoring Instrument (TROPOMI) (Veefkind et al., 2012), launched in Oct. 2017, extends these observation records. The TROPOMI instrument has a small pixel size of 3.6 km across-track by 7.2 km along-track at nadir and provides detailed daily global $NO_2$ images. In August 2019, TROPOMI was switched to smaller pixel size of 3.6 km × 5.6 km.

In TROPOMI tropospheric $NO_2$ retrievals, the stratospheric $NO_2$ has to be subtracted from the total $NO_2$ column. Several approaches have been developed to separate the stratospheric $NO_2$ and tropospheric $NO_2$ (e.g., Richter and Burrows, 2002; Bucsela et al., 2006; Beirle et al., 2016). In the KNMI $NO_2$ algorithm, the stratospheric $NO_2$ is simulated through the assimilation of the TROPOMI $NO_2$ slant columns in the TM5-MP model (van Geffen et al., 2019).

Validation of satellite $NO_2$ products has been done with ground-based measurements over land. For the TROPOMI products, there is the routine validation in the sentinel-5P mission performance centre (http://mpc-vdaf.tropomi.eu/). OMI stratospheric $NO_2$ product has been evaluated by Belmonte-Rivas et al. (2014) and Dirksen et al. (2011). Validation of satellite-based $NO_2$ measurements over oceans using shipborne MAX-DOAS measurements are not routine. Few shipborne MAX-DOAS measurements have been used for the validation of SCIAMACHY and GOME-2 trace gas products (e.g., Krueger and Quack, 2013; Peters et al., 2012; Behrens et al., 2019). Research cruises usually follow routes different from commercial ships: these routes are mostly across remote oceans where there is little or no pollution in the troposphere. Therefore, the ship cruises provide a good opportunity for measuring background $NO_2$ concentration.

From December 2017 to June 2019, we had four opportunities to participate in ship cruises with a MAX-DOAS instrument on-board the German research vessel Sonne and one cruise on-board the German research vessel Maria S. Merian. Four of the cruises were transit cruises, and therefore our measurements covered a large latitude and longitude range, thus providing measurements of latitude gradients in $NO_2$ vertical columns. The cruises are listed in Table 1 and shown in Fig. 1. During transit cruises, the ship usually sails continuously at about $22\,\mathrm{km\,h^{-1}}$ with only a few short stops for activities, such as deployment of Argo floats, while during normal campaign cruises the ship may stay stationary at one or two locations for some days. Because the ship sails over remote oceans, we mainly measured the background tropospheric $NO_2$ and the stratospheric $NO_2$.

In this paper we show the results of the MAX-DOAS measurements during the five cruises and compare the MAX-DOAS measurements with the TROPOMI measurements and TM5-MP model simulations. This paper has the following structure:





Section 2 describes the data sets used in the paper, Section 3 describes the data analysis method, the results and some discussions are shown in Section 4, and Section 5 presents the conclusions.

## 2 Data sets

### 2.1 Data from ship cruises

This section describes ship-based data sets used in this paper, i.e.: the scientific data sets of the MAX-DOAS and Microtops, as well as data measured by the ship's instruments (GPS system and automatic weather station).

#### 2.1.1 Ship cruises and weather data

The R.V. Sonne and R.V. Maria S. Merian provide extensive position and ship state data as well as weather station data at high time resolution during the cruises. The data sets include time, latitude, longitude, and course from the ship's GPS, and heading, pitch and roll of the ship from its compass and inertial systems. The weather data consists of wind speed, wind direction, air temperature, pressure, relative humidity, water temperature, short wave and long wave radiation. The short wave and long wave radiation are only measured outside of the exclusive economic zones (EEZ) of the countries that the ship sailed through. The time, latitude and longitude are important to obtain an accurate ship position and calculate the local solar zenith angle. The heading is used to calculate the viewing azimuth angle of the MAX-DOAS instruments. We downloaded the ship data at 1 minute time resolution.

The ships were quite stable measurement platforms, with pitch and roll values mainly within $\pm 1°$ during the cruises. For most of the cruises, the relative wind direction was mostly from the front of the ship. However, in cases where the relative wind direction was from the stern (back) of the ship, there was a risk that the exhaust gases of the ship's smoke stack came into the field of view of the MAX-DOAS, which could contaminate the measurement. The ship speed was usually $22 \, \mathrm{km \, h^{-1}}$ during the transit cruises. Cruises with a oceanographic purpose had more stationary time. An example is Sonne cruise SO268-1 in March 2019, which was mainly stationary at two locations in the Pacific Ocean. The air temperature in the tropical regions ranged mainly between 25 and 30 °C. There were a few cloud-free days, but most days were partly cloudy. During the trip at the end of 2017 there were five days (25-29 Dec. 2017) with very heavy aerosols in the Atlantic Ocean near the coast of Mauritania (latitude from 25° N to 5° S). There were also several days with rain during the cruises.

#### 2.1.2 MAX-DOAS data

Two similar compact Airyx MAX-DOAS instruments have been used in the cruises. One MAX-DOAS instrument was used in the cruise on-board R.V. Sonne from December 2017 to January 2018. Another MAX-DOAS instrument was used in four cruises, the R.V. Maria S. Merian (MSM for short hereafter) cruise in December 2018 and three Sonne cruises in 2019. The compact MAX-DOAS instrument contains of an Avantes spectrometer, a scanning mirror, a computer, a web camera, and



a GPS. Similar instruments have been used in the Cabauw Intercomparison of Nitrogen Dioxide Measuring Instruments 2 (CINDI2) campaign (Kreher et al., 2019).

The MAX-DOAS instrument was mounted on the railing of the observation deck of Sonne at the same position during the four Sonne cruises. During the Sonne cruise in December 2017 and January 2018, the instrument was pointed to 200° with respect to the ship forward direction. On the MSM, the MAX-DOAS was installed below the observation deck behind the bridge of the ship and pointed 90° with respect to the ship forward direction. During the Sonne cruises in 2019, the MAX-DOAS instrument was pointed to 180° to the ship forward direction. In March and June 2019, the MAX-DOAS was on the Sonne cruises without a KNMI scientist on-board.

The MAX-DOAS performed measurements in both forward and backward directions with respect to the instrument itself. When the solar zenith angle (SZA) was smaller than 84°, the instrument scanned at elevation angles (i.e. the viewing angle above the horizon) of 15° and 30° in forward direction, 90° (zenith), and 150° and 165° in backward direction. During the Sonne cruises in 2019, the 8° and 172° elevation angles were added to the scanning series. The measurement time was about 1 minute per elevation angle.

When the solar zenith angle was between 84° and 97°, the MAX-DOAS performed zenith measurements (90° elevation angle) only. When the SZA was greater than 100°, MAX-DOAS performed dark current and offset measurements. The dark current and offset measurements are used to check the stability of the instruments. The computer time was synchronized to the GPS time at the start of the measurements in the morning. The SZA was calculated by the MAX-DOAS operation software using the computer time and the position of the ship.

The temperature of the spectrometer was stabilized at 20 °C during the trips. The telescope has a heating unit to prevent ice but the temperature of the telescope is not stabilized. During the cruises, MAX-DOAS performed measurements automatically every day, except for the days sailing inside the EEZ. Sometimes MAX-DOAS measured the emissions from the ship itself but these data were not used in this paper.

### 2.1.3 Aerosol data

Aerosol data were measured using a hand-held Microtops sun-photometer (Smirnov et al., 2009). The measurements were performed manually by pointing the sun-photometer to the sun when there were no clouds in the viewing direction of the sun, roughly every 20 minutes. The Microtops measures aerosol optical thickness (AOT) at five wavelengths and total water vapour column. The Angstrom coefficients are calculated from the AOTs. The data derived from the Microtops directly are called level 1 data which are sent to NASA Maritime Aerosol Network (MAN) for cloud screening and quality control. This process generates Microtops level 1.5 and level 2 data, which we downloaded from the NASA MAN website after the cruises. These Microtops data include daily time series and daily mean for AOTs, Angstrom coefficients, and total water vapour column.

The daily aerosol optical thickness time series data were used in the MAX-DOAS data analysis. For each day, the AOT time series were interpolated at the MAX-DOAS measurement time. On the days without aerosol data, an AOT of 0.05 was used in the data analysis. The Microtops daily mean AOT at 500 nm is shown in Fig. 1. During the cruise SO259-3 in December 2017, the ship entered a dust plume on 25 December 2017 at 25° N and sailed out of the dust plume on 30 December 2017 at 5° S.





In this region, the aerosol optical thickness increased from 0.05 to 0.7 on 25 December 2017. The largest AOT was about 1.5; the AOT was $\geq 1$ for three days when the visibility was a few hundred meters and the ship was covered by dust. During the other cruises the AOT values were low, about 0.1 or less at 500 nm, mainly due to sea salt aerosols. The lowest AOT value was about 0.03 at 500 nm during one of the cruises.

## 2.2 TROPOMI data

The TROPOMI $NO_2$ product was developed at KNMI and is generated within the TROPOMI ground segment (PDGS) operational at the German Aerospace Centre (DLR) (van Geffen et al., 2019). The TROPOMI $NO_2$ product provides tropospheric, stratospheric, and total vertical column densities (VCDs), as well as some detailed results.

The KNMI TROPOMI $NO_2$ retrieval algorithm is based on a retrieval/data-assimilation system, following the approach introduced for the OMI $NO_2$ retrievals (the DOMINO approach) (Boersma et al., 2007, 2011) and also applied for the OMI retrievals within the QA4ECV project (Boersma et al., 2018). The total $NO_2$ slant columns are derived using the Differential Optical Absorption Spectroscopy (DOAS) method (Platt and Stutz, 2008). Then the total slant columns are assimilated in the TM5-MP model to determine the stratospheric $NO_2$ slant columns. The tropospheric $NO_2$ slant column is the total slant column minus the stratospheric slant column, after which these slant columns are converted to the tropospheric and stratospheric $NO_2$ VCDs using appropriate air mass factors (AMFs).

The TROPOMI overpass is at about 13:30 local time. On any given day the TROPOMI measurement closest in space and time to one of the MAX-DOAS measurements was selected as the overpass pixel. The mean and standard deviation of the $3 \times 3$ and $5 \times 5$ pixels around the overpass pixel were also determined. TROPOMI data was not available for the cruise from December 2017 to January 2018 when the instrument was still in its in-orbit test phase. Only data with Quality Assurance (QA) value of $> 0.75$ (i.e. cloud radiance fraction $< 0.5$) were selected.

## 2.3 TM5-MP model data

The baseline method in the TROPOMI $NO_2$ algorithm to separate stratospheric and tropospheric contributions to the $NO_2$ total slant columns is by data assimilation of slant columns in the TM5-MP chemistry transport model (Huijnen et al., 2010; Williams et al., 2017). The TM5-MP $NO_2$ profiles are simulated globally at $1° \times 1°$ (latitude x longitude) grids at 35 levels from surface to about 0.01 hPa. The time interval of the output is 30 minutes. The TM5-MP $NO_2$ profiles are kept in archive at KNMI. We selected the $NO_2$ profiles along the ship tracks every day. The number of grid cells from the TM5-MP model collocated with the ship in space and time varied from 1 to 6 per day, depending on the speed of the ship and its activities. The total, stratospheric, and tropospheric $NO_2$ vertical columns were integrated using the TM5-MP $NO_2$ profiles. The tropopause level provided in the TM5-MP data was used to separate the stratospheric and tropospheric $NO_2$ columns. The collocated TM5-MP data are available for four cruises. There are no TROPOMI $NO_2$ data for the first cruise, therefore also no TM5-MP data.





## 3 Data analysis for MAX-DOAS

### 3.1 Fitting of $NO_2$ slant columns

The $NO_2$ slant columns were retrieved with the DOAS technique (Platt and Stutz, 2008) using software developed at KNMI.
The MAX-DOAS spectra were corrected for the dark current and offset measured on the same day. For some days without the dark current and offset spectra measurements, the dark current and offset spectra from nearby days were used. Wavelength calibration was performed using the measurement at the 15° elevation angle in every measurement series. The full width half maximum (FWHM) of the instrument spectral response function was fitted during the wavelength calibration. The FWHM is about 0.6 nm for the MAX-DOAS instruments.

For the DOAS fit we used the settings commonly used in the MAX-DOAS community (e.g., Piters et al., 2012; Kreher et al., 2019). The fitting window was 425-490 nm. For the stratospheric $NO_2$ fit, the cross sections included were $NO_2$ at 220 K (Vandaele et al., 1998), $O_3$ at 223 K (Bogumil et al., 2003), water vapour (Rothman et al., 2010), $O_2$-$O_2$ (Hermans et al., 2001), Ring cross section based on a solar spectrum from Kurucz et al. (1984). For the tropospheric $NO_2$ fit, the $O_3$, water vapour, $O_2$-$O_2$, and Ring cross sections were the same as those used in the stratospheric $NO_2$ fit but the $NO_2$ cross section at
298 K (Vandaele et al., 1998) and the $NO_2$ cross section at 220 K which was made orthogonal with the 298 K cross section were used. A fifth order polynomial of the wavelength was also included in the fits.

    In the DOAS fit, one removes the solar Fraunhofer lines by using the ratio of the measured spectrum and a reference spectrum. Because both spectra are influenced by the instrument spectral response function, the solar Fraunhofer lines cannot be removed completely in the ratio. Since this effect comes from the solar spectrum $I_0$, it is referred to as "$I_0$ effect". Detailed
explanation and corrections for the $I_0$ effect was presented by Alliwell et al. (2002). The $NO_2$ and $O_3$ cross sections have been corrected for the $I_0$ effect.

    For the fit of tropospheric $NO_2$, the reference spectrum was the measurement at 90° elevation angle (zenith) at every scanning series. For the stratospheric $NO_2$, the reference spectrum for the MAX-DOAS measurements from December 2017 to January 2018 was taken on 3 January 2018. The reference spectrum for the MAX-DOAS measurements in December 2018 and 2019
was taken on 3 February 2019. These two reference spectra were measured at solar zenith angle 17° and 24° in the afternoon, at 90° elevation angle, and in cloud free situations. We did not use spectra measured at solar zenith angle close to 0° because of saturation of the detector.

### 3.2 Computation of $NO_2$ vertical columns

The $NO_2$ slant columns present the amount of $NO_2$ along the effective light path from the sun to the MAX-DOAS. In order to
180 convert the slant columns to the vertical columns, air mass factors (AMFs) were calculated using the Doubling-Adding KNMI radiative transfer codes (DAK) (De Haan et al., 1987; Stammes, 2001), with a pseudo-spherical correction (because of the large solar zenith angles up to 89°) and tropical atmospheric profiles of temperature and pressure (Anderson et al., 1986). The $NO_2$ profile was taken from the TM5-MP model simulations and interpolated at the tropical atmospheric profile levels. For the stratospheric AMF, the tropospheric $NO_2$ mixing ratio was set to zero at the altitude from 0 to 18 km, which is about the





tropopause height from the model for the tropical regions. The $NO_2$ total column in the tropical atmospheric profile is about $2.0\times10^{15}$ molec cm$^{-2}$. $NO_2$ photolysis at twilight was not taken into account in the AMF calculations. The uncertainty of the AMFs caused by the neglecting of the $NO_2$ photolysis will be discussed later. Aerosols were specified in a well-mixed layer from 0 to 1 km with aerosol optical thickness values from 0 to 2 in 20 intervals. A Henyey-Greenstein phase function was used for aerosols in the computations.

AMFs for the stratospheric and tropospheric $NO_2$ were calculated off-line separately and stored in look-up tables. The AMF is a function of elevation angle, solar zenith angle, relative azimuth angle, aerosol optical thickness, surface albedo and surface height. For the ship measurements, we set the surface albedo to 0.05 and the surface height to 0 km. The solar zenith angles ranged from 0° to 89°. The AMFs were calculated at the wavelength of 460 nm. The method for the calculation of the tropospheric AMFs is described by Vlemmix et al. (2010).

Clouds were not taken into account in the AMF computations. According to Van Roozendael and Hendrick (2012) clouds are not important for the stratospheric $NO_2$ retrievals using MAX-DOAS. The impact of clouds on tropospheric $NO_2$ retrievals has been analysed by Vlemmix et al. (2015), by analysing the fully cloudy scenes (both zenith and off-axis elevation having clouds) and partly cloudy scenes (one elevation having clouds, either zenith or off-axis). They have reported that for the fully cloudy scenes, the impact of clouds on the sensitivity of MAX-DOAS tropospheric $NO_2$ measurement is small. For the partly

cloudy scenes, the clouds have strong impact on the MAX-DOAS tropospheric $NO_2$ measurements, but the impact can be reduced if the MAX-DOAS data are averaged in time.

The viewing azimuth angles of the MAX-DOAS measurements were corrected using the heading data of the ship. The elevation angles were not explicitly corrected for the pitch and roll of the ship in our calculations because the MAX-DOAS instruments had an automatic continuous adjustment of the elevation angles during the measurements. Because we use 15°

(165°) and 30° (150°) elevation angles in the $NO_2$ retrievals, the 1 degree of pitch and roll are not important for these elevation angles. The solar zenith angles and relative azimuth angles have been re-computed using the ship GPS data because the internal GPS of the MAX-DOAS instrument was malfunctioning.

The stratospheric $NO_2$ vertical columns ($VCD_{strat}$) are calculated using Eq. 1.

$$VCD_{strat} = (DSCD + SCD_{ref})/AMF_{strat} \qquad (1)$$

where DSCD is the differential slant column between the actual slant column and the slant column in the reference spectrum. $SCD_{ref}$ is the slant column in the reference spectrum which is calculated using the total VCD multiplied with the cosine of the SZA. $AMF_{strat}$ is the stratospheric $NO_2$ AMF.

We obtained the total $NO_2$ VCDs in the MAX-DOAS reference spectra from collocated OMI/QA4ECV $NO_2$ data (version 1.1 off-line, at http://www.temis.nl/) (Boersma et al., 2018). The total $NO_2$ column was $1.5\times10^{15}$ molec cm$^{-2}$ in the reference

spectrum on 3 January 2018 and was $1.7 \times 10^{15}$ molec cm$^{-2}$ in the reference spectrum on 3 February 2019.

The tropospheric $NO_2$ vertical columns ($VCD_{trop}$) are calculated using Eq. 2.

$$VCD_{trop} = DSCD_{90}/DAMF \qquad (2)$$





where $DSCD_{90}$ is the differential slant columns between a given elevation angle and $90°$ elevation angle in the same scanning series, and DAMF is the difference between the $NO_2$ AMFs at the given elevation angle and at $90°$ elevation angle.

## 220  4  Results

### 4.1  MAX-DOAS stratospheric $NO_2$

Stratospheric $NO_2$ vertical columns derived from all viewing directions on 5 February 2019 are shown in Fig. 2. On this day, R.V. Sonne sailed at the Pacific ocean ($1.37°$ N - $2.08°$ N, $142.08°$ W - $140.58°$ W). It was good weather with lots of scattered clouds, which was a normal weather condition during the cruises. The stratospheric $NO_2$ VCDs derived from different elevation
angles are quite close to each other but the VCDs are slightly larger at small elevation angles. The $NO_2$ VCD shows a typical diurnal evolution pattern in the stratosphere, with low values in the morning, increasing during the day, and having high values in the evening. These features can be explained by the $NO_x$ related stratospheric chemistry as mentioned in the introduction. The stratospheric $NO_2$ VCD is about $1.5 \times 10^{15}$ molec $cm^{-2}$ at noon and $2.6 \times 10^{15}$ molec $cm^{-2}$ at SZA of $89°$. The values are in the same range as those measured by satellite instruments reported by Belmonte-Rivas et al. (2014).

### 230  4.2  MAX-DOAS tropospheric $NO_2$

Figure 3 shows tropospheric $NO_2$ VCDs on 24 June 2019. The measurement was taken over the Pacific ocean ($25.12°$ N - $24.40°$ N, $137.83°$ E - $134.44°$ E), with scattered clouds. The tropospheric $NO_2$ vertical columns are between 0 and $0.5 \times 10^{15}$ molec $cm^{-2}$ and similar in different elevation angles. So we do not need to separate different elevation angles when comparing MAX-DOAS tropospheric $NO_2$ vertical columns with TROPOMI tropospheric $NO_2$ vertical columns. There is no enhanced
tropospheric $NO_2$ on this day, which is the case for most of the cruises. At SZA larger than $60°$, some tropospheric $NO_2$ VCDs are larger than at noon, which may be the impact of the stratospheric $NO_2$.

As shown in Fig. 3, when the solar zenith angles are larger than $70°$, in the morning the VCDs at the elevations of $150°$ and $165°$ decrease with the increasing SZA; in the evening the VCDs at the elevations of $15°$ and $30°$ decrease with the increasing SZA. The decrease of tropospheric $NO_2$ VCDs with increasing SZA at relatively large SZA is an artefact which is caused
by the rapid changing of the stratospheric $NO_2$ at large SZA and by using the spectrum measured at $90°$ elevation angle as the reference spectrum in every scanning series. The measurements started from the $15°$ elevation angle and finished at the $165°$ elevation angle. In the morning, the spectra at the $150°$ and $165°$ elevation angles are measured later than the reference spectrum and the stratospheric $NO_2$ decreases rapidly in the morning, therefore less $NO_2$ is measured at the $150°$ and $165°$ elevation angles than in the reference spectrum. In the evening, the stratospheric $NO_2$ increases rapidly as SZA increasing and
the spectra at the $15°$ and $30°$ elevation angles are measured earlier than the reference spectrum, consequently, less $NO_2$ is measured at the $15°$ and $30°$ elevation angles than in the reference spectrum. If there is more $NO_2$ in the reference spectrum than in the actual measurement, the DOAS fit may yield a negative $NO_2$ slant column. This artefact has no impact on the





comparison with TROPOMI tropospheric $NO_2$, because the SZAs are small at the TROPOMI overpass time during the four cruises.

## 4.3 Comparison of MAX-DOAS stratospheric $NO_2$ and TM5-MP model simulations

The TM5-MP simulated $NO_2$ profiles were integrated vertically from the tropopause level to the highest level of the model to get the stratospheric $NO_2$ vertical columns. If there were several TM5 latitude/longitude grid cells crossed by the ship in a day, the $NO_2$ columns in the morning (evening) from the first (last) TM5 grid were used to compare with the MAX-DOAS morning (evening) measurements. The SZA values of the TM5 $NO_2$ profiles were calculated at the centre of the latitude and longitude grids. The variation of the $NO_2$ columns in different grid cells crossed by the ship per day was usually small. Figure 4 shows one day of the stratospheric $NO_2$ columns simulated by TM5 and measured by MAX-DOAS on 22 March 2019, when the ship was stationary at the Pacific at $14.5°$ N, $125.5°$ W. On this day the $NO_2$ vertical columns from one TM5 grid cell were selected. The largest SZA in the MAX-DOAS $NO_2$ VCD data is $89°$ in the morning and evening. The MAX-DOAS stratospheric $NO_2$ vertical columns have a similar diurnal variation as the TM5 simulated stratospheric $NO_2$ columns. The TM5 stratospheric $NO_2$ vertical columns have a positive offset compared to the MAX-DOAS stratospheric $NO_2$ VCDs. Plots of other days show a similar pattern.

Figure 5 shows the scatter plot of the MAX-DOAS stratospheric $NO_2$ VCDs measured in the morning and evening versus TM5 simulated stratospheric $NO_2$ VCDs for four cruises. The morning and evening $NO_2$ values are the average of $NO_2$ VCD measured from SZA $75°$ to $89°$, respectively. This solar zenith angle range is used throughout the paper to define the morning and evening $NO_2$. At large SZA, the light path in the stratosphere is longer than that at noon, consequently, the MAX-DOAS measurements are more sensitive to the stratospheric $NO_2$. The MAX-DOAS and TM5 stratospheric $NO_2$ VCDs have a good linear correlation, with a correlation coefficient $R = 0.97$. The mean differences are $3.34 \times 10^{14}$ ($\pm 1.88 \times 10^{14}$) molec cm$^{-2}$ in the morning and $5.69 \times 10^{14}$ ($\pm 3.12 \times 10^{14}$) molec cm$^{-2}$ in the evening. The TM5 stratospheric $NO_2$ VCDs are slightly higher than the MAX-DOAS stratospheric $NO_2$ VCDs. The statistics of the comparisons are summarized in Table 2.

Additionally, we have compared the MAX-DOAS and TM5 stratospheric $NO_2$ VCDs at the SZA ranges of $0°$-$30°$, $30°$-$60°$, $60°$-$75°$. At smaller SZA angles, TM5 simulated stratospheric $NO_2$ VCDs are mostly larger than the MAX-DOAS measurements.

We also compared the TROPOMI stratospheric $NO_2$ VCDs with the TM5-MP model simulated stratospheric $NO_2$ VCDs. They are almost the same, the mean difference is about $-2.49 \times 10^{13}$ molec cm$^{-2}$ (about 1%). This is expected because the TROPOMI $NO_2$ total columns are assimilated in TM5-MP model to separate the stratospheric and tropospheric $NO_2$. This is a good consistency check for the TROPOMI stratospheric $NO_2$ VCDs.





## 4.4 Comparison of MAX-DOAS and TROPOMI NO$_2$ vertical columns

### 4.4.1 Stratospheric NO$_2$

As mentioned before, the MAX-DOAS are more sensitive to the stratospheric NO$_2$ in the morning and evening than at the
TROPOMI overpass time (at 13:30 LT). Because the stratospheric NO$_2$ VCDs have a diurnal cycle, we cannot interpolate
the stratospheric NO$_2$ VCD directly at the TROPOMI overpass time using the MAX-DOAS morning and evening values.
The interpolation has to be done using a chemistry model as presented by Tack et al. (2015). Since the TM5 and MAX-DOAS
stratospheric NO$_2$ VCDs had a similar diurnal cycle, we used the TM5 model to interpolate the MAX-DOAS stratospheric NO$_2$
VCDs at the TROPOMI overpass time. First, for each day the TM5 stratospheric NO$_2$ VCDs were shifted to the MAX-DOAS
stratospheric NO$_2$ VCDs by subtracting the mean difference of the stratospheric NO$_2$ VCDs between TM5 and MAX-DOAS
for SZA between 75 and 89°. The stratospheric NO$_2$ VCD at the TROPOMI overpass time (called TM5 interpolated NO$_2$
VCD) was interpolated using this corrected (shifted) TM5 stratospheric NO$_2$ VCDs.

The stratospheric NO$_2$ VCDs of MAX-DOAS and TROPOMI for the cruise in February 2019 are shown in Fig. 6. The figure
shows the MAX-DOAS stratospheric NO$_2$ VCDs collocated with the TROPOMI measurements, the MAX-DOAS morning
and evening stratospheric NO$_2$ VCDs, and the TM5 interpolated stratospheric NO$_2$ VCDs. In absolute terms, the MAX-DOAS
stratospheric NO$_2$ VCDs are smaller in the morning and larger in the evening. The MAX-DOAS NO$_2$ VCDs collocated to
the TROPOMI overpass and the interpolated TM5 stratospheric NO$_2$ VCDs are between the morning and evening values. The
MAX-DOAS NO$_2$ VCDs are lower than the TROPOMI NO$_2$ VCDs. In some cases, there was no TROPOMI data due to the
presence of clouds (with a cut off at a cloud radiance fractions of 0.5).

A scatter plot of TROPOMI versus MAX-DOAS stratospheric NO$_2$ VCDs for all the cruises is shown in Fig. 7. The
TROPOMI values were taken from the pixels collocated to the MAX-DOAS location. If the collocated MAX-DOAS NO$_2$
measurement was contaminated by ship emissions, then the NO$_2$ VCD was derived from unpolluted data within 7 minutes
around the overpass time. We removed the MAX-DOAS data measured on the days when the wind direction was from the back
of ship and the exhaust of the ship was measured. On the days when the wind was from the front of the ship, MAX-DOAS
sometimes measured a small amount of exhaust NO$_2$ for a few minutes; these peaks were also removed. If the collocated
MAX-DOAS NO$_2$ was larger than the MAX-DOAS NO$_2$ VCD at the SZA of 80° in the evening, the collocated MAX-DOAS
NO$_2$ VCD was flagged as polluted. For the data in Fig. 7, the correlation coefficient is 0.93, mean difference of $2.42 \times 10^{14}$
molec cm$^{-2}$ and standard deviation of $2.24 \times 10^{14}$ molec cm$^{-2}$. The linear fit of the TROPOMI and MAX-DOAS stratospheric
NO$_2$ VCDs has a slope of 1.076 and an offset of $0.74 \times 10^{14}$ molec cm$^{-2}$.

Figure 8 shows the TROPOMI stratospheric NO$_2$ VCDs versus the TM5 interpolated stratospheric NO$_2$ VCDs. The correla-
tion coefficient is 0.95 with a mean difference of $4.34 \times 10^{14}$ molec cm$^{-2}$ and a standard deviation of $1.92 \times 10^{14}$ molec cm$^{-2}$.
The linear fit of the TROPOMI and TM5 interpolated stratospheric NO$_2$ VCDs has a slope of 1.083 and an offset of $2.653 \times 10^{14}$
molec cm$^{-2}$, which is similar to that of Fig. 7.

The MAX-DOAS and TROPOMI stratospheric NO$_2$ VCDs for all cruises are shown as a function of latitude in Fig. 9. Both
data sets illustrate the latitudinal dependency of the stratospheric NO$_2$ VCDs, with low values in tropical region (20° S to 10°





N) and higher values at mid-latitudes (10° N - 40° N). Note that the MAX-DOAS data were taken in four cruise in different months, not in a single cruise. The latitudinal dependency is well-known in satellite stratospheric $NO_2$ VCD data (Belmonte-Rivas et al., 2014). In the tropics the low stratospheric $NO_2$ VCDs are caused by upward and poleward transport in the Hadley cell (Noxon, 1979).

### 4.4.2 Tropospheric $NO_2$

The tropospheric $NO_2$ VCDs for the cruise in February 2019 across the Pacific is shown in Fig. 10. There are no anomalous high tropospheric $NO_2$ VCDs during this cruise. As shown in the figure, most MAX-DOAS tropospheric $NO_2$ VCDs are close to zero. And the TROPOMI tropospheric $NO_2$ VCDs are also very low, $7 \times 10^{14}$ molec cm$^{-2}$, with large error bars because of the low $NO_2$ concentrations (van Geffen et al., 2019).

Figure 11 shows the scatter plot of TROPOMI tropospheric $NO_2$ VCD versus MAX-DOAS tropospheric $NO_2$ VCD at the closest overpass time. The vertical error bar is the uncertainty of the TROPOMI tropospheric $NO_2$ VCD, which is taken from the TROPOMI data. The horizontal error bar is for the MAX-DOAS tropospheric $NO_2$ VCD, which is assumed to be 100% of the $NO_2$ VCD. We can see that the MAX-DOAS and TROPOMI data both show low tropospheric $NO_2$ during these cruises. The TROPOMI and MAX-DOAS tropospheric $NO_2$ VCDs are in the same range, most of the points are between 0 and $5 \times 10^{14}$ molec cm$^{-2}$. Because of very low tropospheric $NO_2$, there is almost no correlation between the tropospheric $NO_2$ VCDs. The mean difference and standard deviation are $4.00 \times 10^{14}$ and $5.08 \times 10^{14}$ molec cm$^{-2}$, respectively.

The negative values in the MAX-DOAS tropospheric $NO_2$ are mostly due to the low $NO_2$ values and the detection limit of the MAX-DOAS. The negative tropospheric $NO_2$ VCD values may also be caused by the clouds in the reference spectrum but not in the off-axis spectrum. The best root mean square error in the DOAS fit for tropospheric $NO_2$ is $1.2 \times 10^{-4}$. The $NO_2$ cross section is about $1 \times 10^{-19}$ cm$^2$ molec$^{-1}$. If we assume that twice of the RMS can be detected, the detection limit for the slant column is $2.4 \times 10^{15}$ molec cm$^{-2}$. The AMF for the 15° elevation angle is about 2.2, hence the detection limit for the vertical column is $1.1 \times 10^{15}$ molec cm$^{-2}$. This estimation of the detection limit is similar to that used by Peters et al. (2012). They proposed this value as an upper limit, the actual detection limit can be better than this. During the cruises, tropospheric $NO_2$ slant columns larger than $2.4 \times 10^{15}$ molec cm$^{-2}$ were rarely detected.

### 4.5 Discussions

Because the reference spectra were measured by the MAX-DOAS during the cruises, there was background $NO_2$ absorption in the reference spectra. The $NO_2$ VCD in the reference spectrum was estimated using the collocated OMI/QA4ECV $NO_2$ VCD, which may cause an uncertainty (offset) in the MAX-DOAS stratospheric $NO_2$ VCDs. Zara et al. (2018) reported that the uncertainty of the OMI $NO_2$ SCD in remote ocean region was about $8 \times 10^{14}$ molec cm$^{-2}$. The uncertainty of the $NO_2$ VCD in the reference spectrum is estimated to be $4 \times 10^{14}$ molec cm$^{-2}$ because the AMF is about 2 at noon. The $NO_2$ VCD in the reference spectrum has a larger impact on the stratospheric $NO_2$ VCD at the TROPOMI overpass time, for example in the comparison of MAX-DOAS $NO_2$ VCD with TROPOMI at the collocated pixels. Since the same reference spectrum is used for the MAX-DOAS analysis, the impact of the reference spectrum on the MAX-DOAS stratospheric $NO_2$ VCD is the same





for all trips. The $NO_2$ in the reference spectrum has less impact on the MAX-DOAS stratospheric $NO_2$ VCD at the SZA range

of 75°-89°, because the mean AMF in this SZA range is about 7 time the AMFs of the reference spectrum (due to the long light path at large SZAs).

Neglecting the $NO_2$ photo-dissociation may lead to 10% uncertainty in the AMFs at twilight because of the change of the $NO_2$ profiles (Van Roozendael and Hendrick, 2012). Since we only used the measurements at SZA smaller than 89°, the impact from the photo-dissociation may be smaller in our analysis. We have calculated the stratospheric $NO_2$ AMFs using a range of

350 $NO_2$ profiles from the TM5 output. The AMFs for the stratospheric $NO_2$ are very similar and the differences are within 5%.

In the DOAS fit, the uncertainty of the MAX-DOAS stratospheric $NO_2$ slant columns is about $0.5 \times 10^{14}$ molec cm$^{-2}$ at SZA of 20° and increases to $1 \times 10^{14}$ molec cm$^{-2}$ at SZA of 80°. These uncertainties are given in the output of our DOAS fit program. The uncertainty of the $NO_2$ VCD in the reference spectra is about $4 \times 10^{14}$ molec cm$^{-2}$ based on the OMI data. The stratospheric $NO_2$ AMFs are about 1.2 and 5.5 at 20° and 80° of the SZA with an uncertainty of 10%. Using the uncertainty

estimation method presented by Tack et al. (2015), in total, we estimate that the uncertainty of the stratospheric $NO_2$ VCD is about $4 \times 10^{14}$ molec cm$^{-2}$ and $1 \times 10^{14}$ molec cm$^{-2}$ at SZA of 20° and 80°, respectively.

For the tropospheric $NO_2$ VCDs, assuming the AMF of 2.0 with an uncertainty of 10%, the uncertainty of the tropospheric $NO_2$ VCD is estimated to be $2.1 \times 10^{14}$ molec cm$^{-2}$. However, Bais et al. (2016) recommended that the $NO_2$ differential AMF uncertainties to be used for MAX-DOAS at 15° and 30° elevations were 41% and 22%, respectively. In reality the uncertainty

of the MAX-DOAS tropospheric $NO_2$ VCDs is larger than the values given here.

The comparison of MAX-DOAS and TROPOMI stratospheric $NO_2$ VCDs has also been analysed using averaged TROPOMI data over $3 \times 3$ and $5 \times 5$ ground pixels around the collocated pixels. The mean differences between TROPOMI and MAX-DOAS stratospheric $NO_2$ VCDs are 4.34, 4.57, 4.55 $\times 10^{14}$ molec cm$^{-2}$ for 1, $3 \times 3$, and $5 \times 5$ pixels, respectively. The best agreement between the TROPOMI and MAX-DOAS stratospheric $NO_2$ VCDs occurs for the single pixel cases presented in

this paper.

The comparisons of TROPOMI stratospheric $NO_2$ VCDs with MAX-DOAS collocated stratospheric $NO_2$ VCD and with the TM5-MP interpolated stratospheric $NO_2$ VCDs show consistent results: TROPOMI stratospheric $NO_2$ VCDs are higher than the other two products. The TROPOMI stratospheric $NO_2$ VCDs have good linear correlation with the MAX-DOAS collocated and TM5 interpolated stratospheric $NO_2$ VCDs. The linear fit of the TROPOMI stratospheric $NO_2$ VCDs and MAX-DOAS

collocated stratospheric $NO_2$ VCDs or TM5 interpolated stratospheric $NO_2$ VCDs have similar slopes and offsets.

The differences of the MAX-DOAS and TROPOMI $NO_2$ VCDs do not depend on the cloud radiance fraction. The MAX-DOAS tropospheric $NO_2$ VCDs are close to the detection limit. The negative values can also be due to clouds observed in the 90° elevation angle but not in the off-axis elevation angle. These MAX-DOAS tropospheric $NO_2$ VCDs provide an evaluation of the lowest TROPOMI tropospheric $NO_2$ values; such clean cases are not easily observed over land.





# 5   Conclusions

We have presented MAX-DOAS measurements during five cruises from 2017 to 2019, covering a large latitude and longitude range, in both summer and winter. The MAX-DOAS measurements have been compared with TROPOMI stratospheric and tropospheric $NO_2$ vertical columns. Since the TM5-MP model is used in the TROPOMI retrievals, we also compared MAX-DOAS $NO_2$ VCDs with the TM5-MP simulations. It turns out that TROPOMI stratospheric $NO_2$ vertical columns have a

good linear correlation with MAX-DOAS stratospheric $NO_2$ vertical columns. Compared to the MAX-DOAS measurements, TROPOMI has a small positive bias of 2.4 to $4.3 \times 10^{14}$ molec cm$^{-2}$ (10-20%), with an uncertainty of $2 \times 10^{14}$ molec cm$^{-2}$. The uncertainty of MAX-DOAS stratospheric $NO_2$ vertical columns is estimated to be 1 to $4 \times 10^{14}$ molec cm$^{-2}$.

Because the cruises were mostly in remote ocean areas, the MAX-DOAS tropospheric $NO_2$ values were quite low, often close to 0 or slightly negative as a result of low detection limit or impact of clouds. The mean of the collocated TROPOMI

tropospheric $NO_2$ VCDs is $4.7 \times 10^{14}$ molec cm$^{-2}$. The mean difference between TROPOMI and MAX-DOAS $NO_2$ VCDs is $4.0 \times 10^{14}$ molec cm$^{-2}$ with a standard deviation of $5.1 \times 10^{14}$ molec cm$^{-2}$. The uncertainty of MAX-DOAS tropospheric $NO_2$ vertical columns is about $2 \times 10^{14}$ molec cm$^{-2}$. We can confirm that both TROPOMI and MAX-DOAS measured very low tropospheric $NO_2$ VCDs over clean oceans.

*Data availability.*   MAX-DOAS data are available from the authors, TROPOMI data are available from the Copernicus website.

*Competing interests.*   No competing interests are present.

*Acknowledgements.*   We would like to thank Dr. Stefan Kinne (MPI-M) for the organization of the cruises and the Microtops measurements, and Dr. Thomas Rutz (FU-Berlin) for taking care of the MAX-DOAS instrument during the cruise in June 2019. We appreciate Dr. Henk Eskes (KNMI) for the discussions about the TM5-MP data. We also want to thank the captains and crews of the German research vessels R.V. Sonne and R.V. Maria S. Merian for their hospitality and support. The support of the Leitstelle Deutsche Forschungsschiffe (German
Research Fleet Coordination Centre) at the University of Hamburg was highly appreciated. The cruises were sponsored/funded by DFG and BMBF in Germany.



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



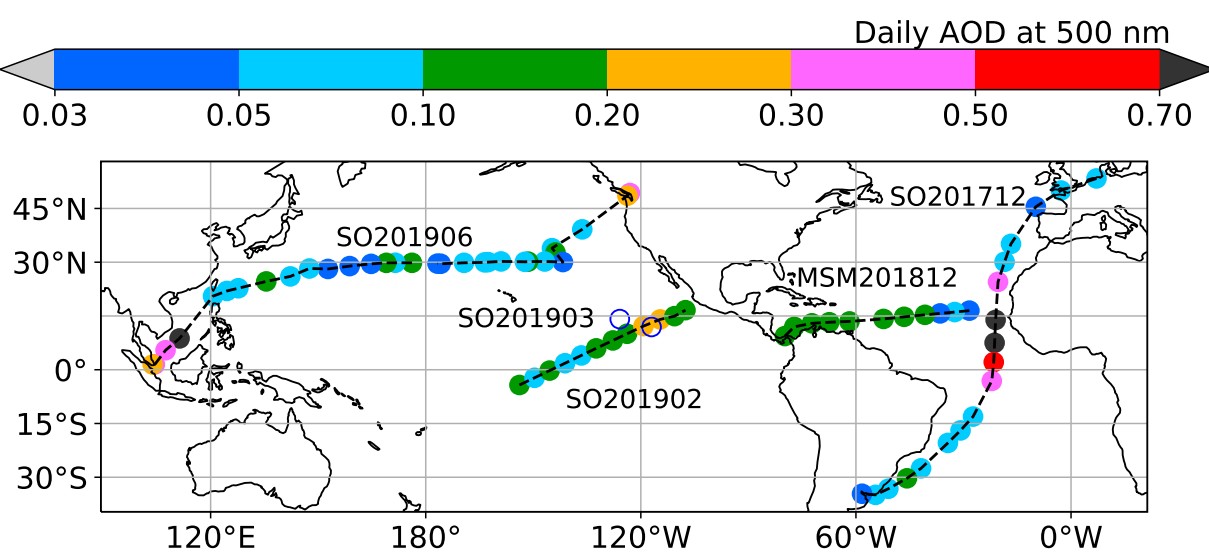

**Figure 1.** Daily aerosol optical thickness at 500 nm (AOT) along the cruise routes. Two open circles indicate the stationary positions of Sonne for the cruise in March 2019 (SO201903), no Microtops measurements.

**Figure 2.** An example of diurnal cycle of stratospheric $NO_2$ VCDs on 5 February 2019. The day fraction is in UTC. 5 Feb. 2019 is day 36. The SZA values are corresponding to the local time. The $NO_2$ VCDs derived from different elevation angles are indicated with different colours. The measurements were taken at the Pacific.





**Figure 3.** An example of one day of tropospheric $NO_2$ VCDs on 24 June 2019. The day fraction is in UTC. 24 June 2019 is day 175. The SZA values are corresponding to the local time. The $NO_2$ VCDs derived from different elevation angles are indicated with different colours. The reference spectrum was taken at the 90° elevation angle in every scan, so no $NO_2$ VCD was retrieved from the 90° elevation angle. The measurements were taken at the Pacific.





**Figure 4.** Stratospheric NO₂ vertical columns on 22 March 2019, measured by MAX-DOAS and simulated by TM5. The day fraction is in UTC. 22 March 2019 is day 81. The SZA values are corresponding to the local time. Sonne was stationary at the Pacific at 14.5° N, 125.5° W. TM5 simulations in one grid were used.



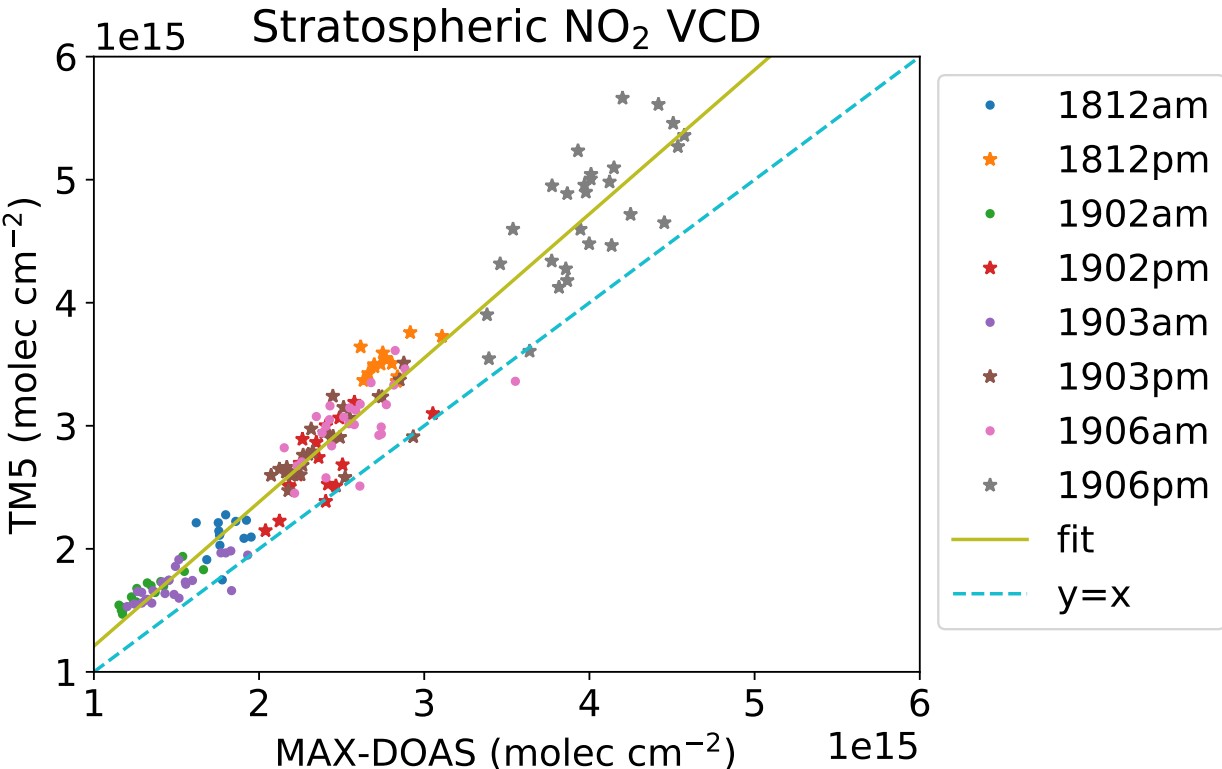

**Figure 5.** Scatter plot of TM5 stratospheric $NO_2$ vertical columns versus MAX-DOAS stratospheric $NO_2$ vertical columns. The am and pm MAX-DOAS $NO_2$ vertical columns are the mean values between solar zenith angle of $75°$ and $89°$ in the morning and in the evening, respectively. The numbers 1812, 1902, 1903, and 1906 refer to the years (2018, 2019) and months (Dec., Feb., Mar., Jun.) of the cruises. The correlation coefficient is R = 0.97. The fit is $y = 1.17x + 3.86 \times 10^{13}$.

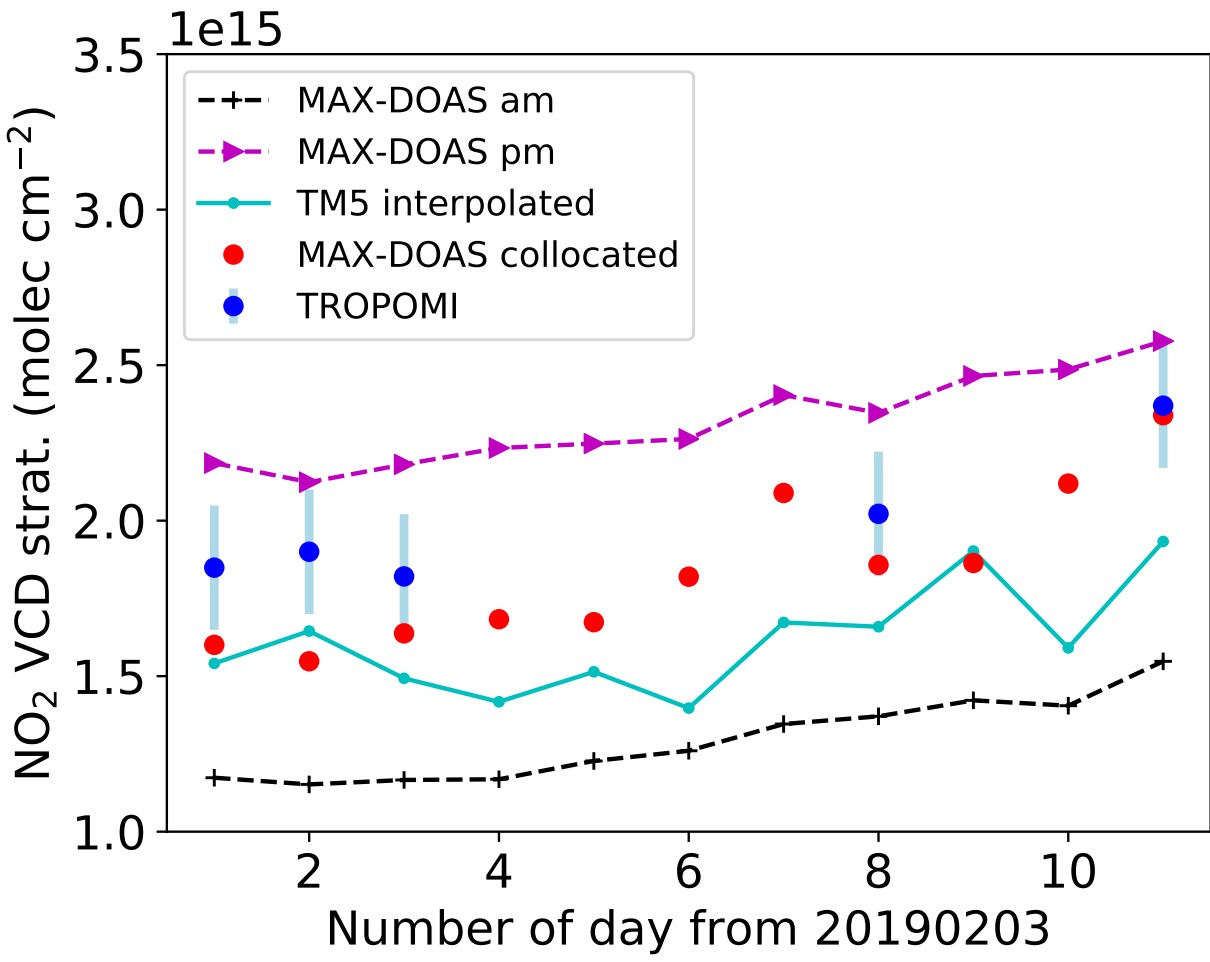

**Figure 6.** Time series of stratospheric $NO_2$ vertical columns for the cruise in February 2019. The MAX-DOAS am and pm $NO_2$ vertical columns are the mean values between solar zenith angle of 75° and 89° in the morning and in the evening, respectively. The missing TROPOMI data are due to clouds.

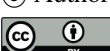

**Figure 7.** Scatter plot of TROPOMI stratospheric $NO_2$ vertical columns versus MAX-DOAS stratospheric $NO_2$ vertical columns. The MAX-DOAS measurements are taken from the collocated TROPOMI pixels. The correlation coefficient is 0.93. The fit is $y = 1.076x + 7.388 \times 10^{13}$.

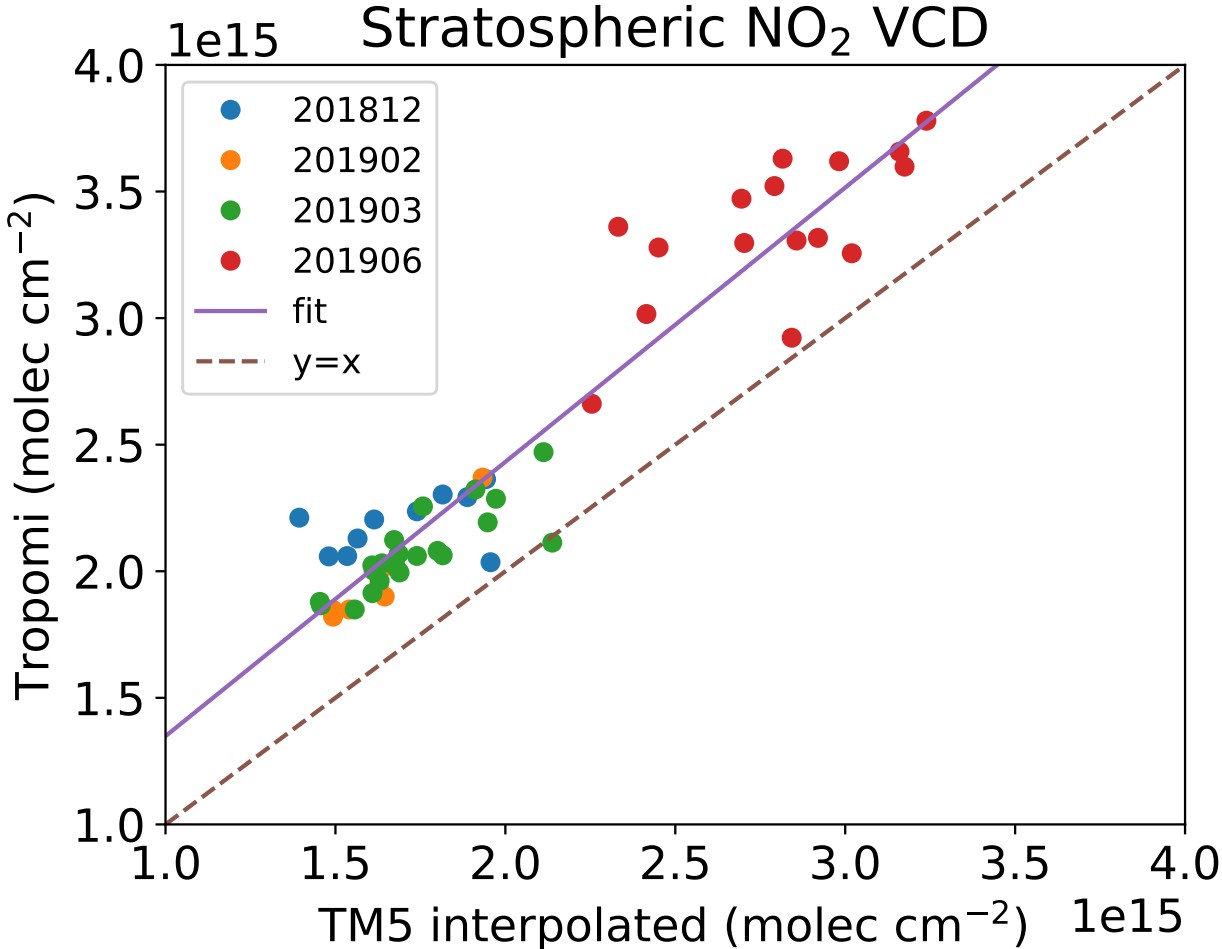

**Figure 8.** Scatter plot of TROPOMI stratospheric $NO_2$ vertical columns versus TM5 interpolated stratospheric $NO_2$ vertical columns with the correction of MAX-DOAS measurements. Same TROPOMI data as in Fig. 7. The correlation coefficient is 0.95. The fit is $y = 1.083x + 2.653 \times 10^{14}$.

**Figure 9.** TROPOMI and MAX-DOAS TM5 interpolated stratospheric NO₂ vertical columns as a function of latitude. Same data as in Fig. 8.

.



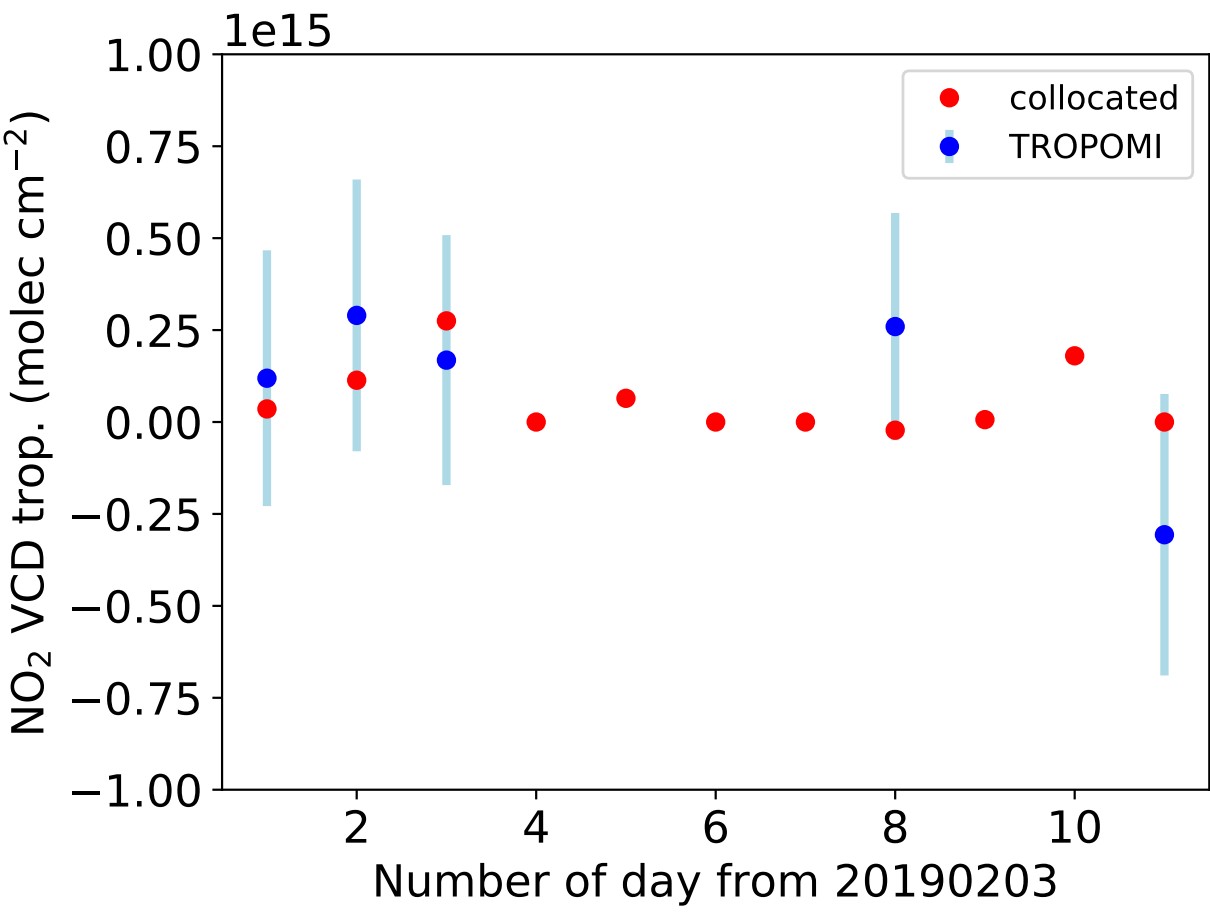

**Figure 10.** Time series of TROPOMI (blue) and collocated MAX-DOAS (red) tropospheric $NO_2$ vertical columns for the cruise in February 2019. The missing TROPOMI data are due to clouds.

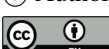

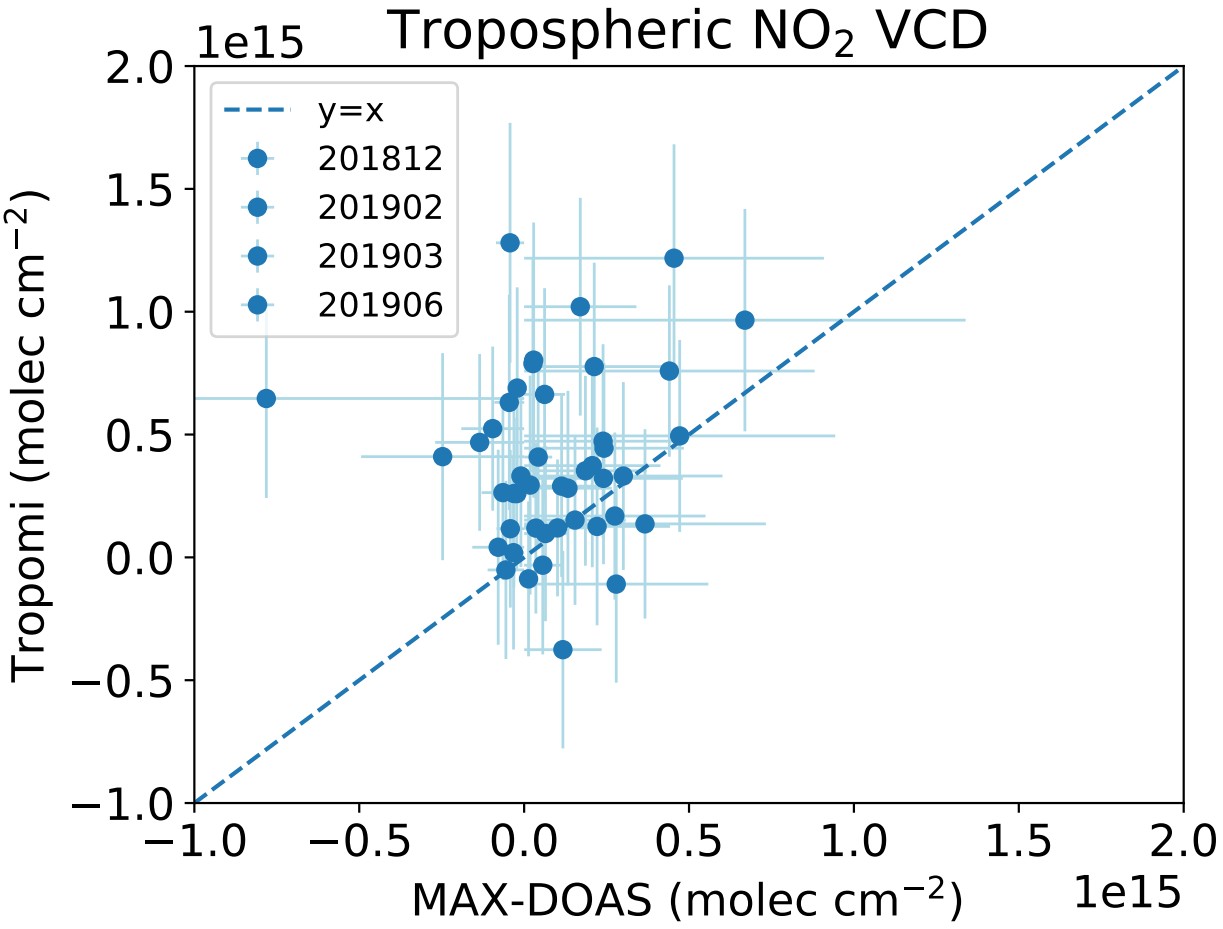

**Figure 11.** Scatter plot of TROPOMI tropospheric NO₂ vertical columns versus MAX-DOAS tropospheric NO₂ vertical columns for all cruises. The MAX-DOAS measurements are taken from the collocated TROPOMI pixels.



**Table 1.** List of R.V. Sonne (SO) and R.V. Maria S. Merian (MSM) cruises with MAX-DOAS measurements

| Number | Cruise | Date | Routes |
|---|---|---|---|
| 1 | SO259-3 / SO201712 | 17 December 2017 - 9 January 2018 | Emden (Germany) - Buenos Aries (Argentine) |
| 2 | MSM79-2 / MSM201812 | 6 December 2018 - 18 December 2018 | Mindelo (Cape Verde) - Bahia de las Minas (Panama) |
| 3 | SO267-2 / SO201902 | 28 January 2019 - 14 February 2019 | Suva (Fiji) - Manzanillo (Mexico) |
| 4 | SO268-1 / SO201903 | 17 February 2019 - 27 March 2019 | Manzanillo (Mexico)- Manzanillo (Mexico) |
| 5 | SO268-3 / SO201906 | 30 May 2019 - 5 July 2019 | Vancouver (Canada) - Singapore |

**Table 2.** Statistic results of the comparison of TROPOMI and MAX-DOAS stratospheric $NO_2$ vertical columns. The MAX-DOAS collocated is the MAX-DOAS stratospheric $NO_2$ VCD collocated with TROPOMI measurement. The TM5 interpolated is the MAX-DOAS stratospheric $NO_2$ VCD interpolated using TM5 stratospheric $NO_2$ diurnal cycle.

| $NO_2$ VCD$_{strat}$ | Mean $\times 10^{15}$ molec cm$^{-2}$ | Standard deviation $\times 10^{15}$ molec cm$^{-2}$ |
|---|---|---|
| TROPOMI (a) | 2.45 | 0.60 |
| MAX-DOAS collocated (b) | 2.21 | 0.52 |
| TM5 interpolated (c) | 2.03 | 0.54 |
| a - b | 0.24 | 0.22 |
| a - c | 0.43 | 0.19 |
| MAX-DOAS $75° \leqslant$ SZA $\leqslant 89°$(d) | 2.42 | 0.88 |
| TM5 $75° \leqslant$ SZA $\leqslant 89°$(e) | 2.87 | 1.05 |
| e - d | 0.45 | 0.28 |