# Peer review of "Shipborne MAX-DOAS measurements for validation of TROPOMI NO2 products"

_Atmospheric Measurement Techniques, 2019_

## Referee Comment (RC1) · Anonymous Referee #1 · 9 Jan 2020

The manuscript describes the comparison between TROPOMI, TM5 model and MAX-DOAS NO2 observations during 5 ship cruises over the Pacific Ocean. The paper is well written and can be published after addressing the following minor comments.

Specific comments

P2 L41 There are a few recent paper on the validation of TROPOMI NO2 over land (some under discussion). Here some examples:

Griffin, D., Zhao, X., McLinden, C. A., Boersma, F., Bourassa, A., Dammers, E., Degenstein, D., Eskes, H., Fehr, L., Fioletov, V., Hayden, K., Kharol, S. K., Li, S.-M., Makar, P., Martin, R. V., Mihele, C., Mittermeier, R. L., Krotkov, N., Sneep, M., Lamsal, L. N., ter Linden, M., van Geffen, J., Veefkind, P., and Wolde, M.: High-

Resolution Mapping of Nitrogen Dioxide With TROPOMI: First Results and Validation Over the Canadian Oil Sands, Geophysical Research Letters, 46, 1049–1060, https://doi.org/10.1029/2018GL081095, 2019.

Ialongo, I., Virta, H., Eskes, H., Hovila, J., and Douros, J.: Comparison of TROPOMI/Sentinel 5 Precursor NO2 observations with ground-based measurements in Helsinki, Atmos. Meas. Tech. Discuss., https://doi.org/10.5194/amt-2019-329, accepted, 2019.

Zhao, X., Griffin, D., Fioletov, V., McLinden, C., Cede, A., Tiefengraber, M., Müller, M., Bognar, K., Strong, K., Boersma, F., Eskes, H., Davies, J., Ogyu, A., and Lee, S. C.: Assessment of the quality of TROPOMI high-spatial-resolution NO2 data products, Atmos. Meas. Tech. Discuss., https://doi.org/10.5194/amt-2019-416, in review, 2019.

P3 L74-76 So I understand you did not use the measurements of ship emissions? I was wondering why; could not be useful to get some of these data for the validation? If you are afraid the resolution of TROPOMI will not be able to detect that I think it's still worth showing. . .

Figure 2 (and all the others) Day fraction: could you use normal time of the day (not decimals)? It's a bit confusing. . .

P9 L267 Could you give these differences also as percentage? (In the abstracts as well)

L282 Could give a brief description of this interpolation method together with the reference? (it remains a bit unclear)

Figures 6,10, 11: What quantity are the error bars? It should be mentioned in the caption

---

## Referee Comment (RC2) · Anonymous Referee #2 · 16 Jan 2020

The manuscript "Shipborne MAX-DOAS measurements for validation of TROPOMI NO$_2$ products" presents results of NO$_2$ MAX-DOAS measurements from several ship cruises in the Atlantic and Pacific Ocean which are used to validate the NO$_2$ measurements of the new TROPOMI instrument and NO$_2$ simulations of the TM5-MP model. They found a good agreement between the three datasets. In general the manuscript is well structured and fits in the scope of AMT. Before publication, the following comments should be considered:

**General comments**:

- Where are the data from the cruise SO201712 shown? Are they only used in Fig. 1? If the data are not used for validation, they should not be shown in this

manuscript.

- discussion of the results of previous studies, for example:

  – P2, L41: What are the results of the validation measurements over land? Please cite some papers.
  – P2, L45: What are the results from the previous studies of the comparison between shipborne and satellite measurements?
  – Sect. 4.5: comparison with previous studies is missing

- The aerosol data are described in Sect. 2.1.3. Is there a relationship between aerosol load and differences between the satellite and MAX-DOAS measurements?

**Specific comments**:

- throughout the manuscript: slant/vertical column → slant/vertical column density.

- throughout the manuscript: "R.V." → "RV"

- P3, L67: absolute or relative wind speed and wind direction?

- P3, L73: "The ships were quite stable measurement platform, ....": Is this really true? For all cruises, pitch and roll angles in a range of $\pm 1°$ sounds for me a little bit unrealistic, especially for the cruise SO201712.

- P3, L79-81: Please move the part of the aerosols to Sect. 2.1.3

- P4, L89-97: The whole set-up of the instrument remains unclear to me. Did the instrument really pointed to the backward? Are the 200° clockwise or counterclockwise? Why are measurements to the back of the ship used? I would assume that these measurements are mostly contaminated by the ship plume.

- P4, L99: What is about the ship movement within one minute? How large is the error which is introduced in the elevation angle?

- P4, L102-103: Maybe it would be good to move this sentence to the description of the position of the instrument.

- P4, L120: Please add also the coordinates in °E.

- P4, L118: Why is an AOT of 0.05 chosen for days without aerosol measurements?

- P5, L129: What are detailed results? Please add some further information.

- P7, L187: What means "later"? Please add an reference.

- P8, L237: Please add a tick in Fig. 3 at 70°? Than it is easier to see.

- P9, L270-272: Can these results be presented in a small table?

- P9, L273-276: The TROPOMI results are unexpected as the section is about MAX-DOAS and TM5-MP. Change title of the section or put the paragraph somewhere else in the manuscript.

- P10, L288: What is with the first days (28 Jan. - 2 Feb.) of the cruise?

- Figure 2: Please use a normal time axis (such as local time), because day fraction is hard to understand. Please add "strat." to y-label and add the coordinates of mean ship position form the 5 Feb. 2019 in the caption.

- Figure 3: Please use a normal time axis (such as local time). Please add "tropo.." to y-label and add the coordinates of mean ship position form the 24 June 2019 in the caption.

- Figure 4: Please use a normal time axis (such as local time).

- Figure 5: Please add a unit to the offset in the figure caption.

- Figure 6: Please adjust the y-label to the label of the other figures. Please use the day of month for the x-axis.

- Figure 7: Please add a unit to the offset in the figure caption. TROPOMI is an acronym. Please use capital letters.

- Figure 8: Please add a unit to the offset in the figure caption. Please change y-label (see Fig. 7).

- Figure 9: What means "MAX-DOAS TM5 interpolated"? In L 309 it is written "...MAX-DOAS and TROPOMI stratospheric $NO_2$...".

- Figure 10: Please add "MAX-DOAS" to the legend. Please adjust y-label (Trop. $NO_2$ VCD) and use day of month for x-axis.

- Figure 11: TROPOMI is an acronym. Please use capital letters in the y-label.

- Table 1: The official cruise names are written with a slash in the name such as SO268/1. Please adjust throughout the manuscript.

---

## Author Comment (AC1) · 20 Feb 2020

*The manuscript describes the comparison between TROPOMI, TM5 model and MAXDOAS NO2 observations during 5 ship cruises over the Pacific Ocean. The paper is well written and can be published after addressing the following minor comments.*

*Specific comments P2 L41 There are a few recent paper on the validation of TROPOMI NO2 over land (some under discussion). Here some examples:*

*Griffin, D., Zhao, X., McLinden, C. A., Boersma, F., Bourassa, A., Dammers, E., Degenstein, D., Eskes, H., Fehr, L., Fioletov, V., Hayden, K., Kharol, S. K., Li, S.M., Makar, P., Martin, R. V., Mihele, C., Mittermeier, R. L., Krotkov, N., Sneep, M., Lamsal, L. N., ter Linden, M., van Geffen, J., Veefkind, P., and Wolde, M.: High Resolution Mapping of Nitrogen Dioxide With TROPOMI: First Results and Validation Over the Canadian Oil Sands, Geophysical Research Letters, 46, 1049–1060, https://doi.org/10.1029/2018GL081095, 2019*

*Ialongo, I., Virta, H., Eskes, H., Hovila, J., and Douros, J.: Comparison of TROPOMI/Sentinel 5 Precursor NO2 observations with ground-based measurements in Helsinki, Atmos. Meas. Tech. Discuss., https://doi.org/10.5194/amt-2019-329, accepted, 2019.*

*Zhao, X., Griffin, D., Fioletov, V., McLinden, C., Cede, A., Tiefengraber, M., Müller, M., Bognar, K., Strong, K., Boersma, F., Eskes, H., Davies, J., Ogyu, A., and Lee, S. C.: Assessment of the quality of TROPOMI high-spatial-resolution NO2 data products, Atmos. Meas. Tech. Discuss., https://doi.org/10.5194/amt-2019-416, in review, 2019.*

Thank you for the references. The papers have been cited in the introduction.

'Validation of TROPOMI satellite NO2 products has been done with ground-based measurements over land (e.g., Griffin et al., 2019; Ialongo et al., 2020; Zhao et al., 2019). '

*P3 L74-76 So I understand you did not use the measurements of ship emissions? I was wondering why; could not be useful to get some of these data for the validation? If you are afraid the resolution of TROPOMI will not be able to detect that I think it's still worth showing...*

The measurements of air that are affected by the ship's emissions are so local (within tens of meters from the smoke stack with a very limited area sampled) that they are in no way representative for the area sampled by the Tropomi pixel. Ships emissions on a well-travelled shipping route show up as a

collective signature but emissions from a single ship on a not well-travelled route disappear in the background.

*Figure 2 (and all the others) Day fraction: could you use normal time of the day (not decimals)? It's a bit confusing…*

We have changed the day fraction on x-axis to local time as suggested by referee #2.

*P9 L267 Could you give these differences also as percentage? (In the abstracts as well)*

We have given the percentage close to L267 and in the abstract, in  lines 275-276, 310 in the revised manuscript.

*L282 Could give a brief description of this interpolation method together with the reference? (it remains a bit unclear)*

We shifted the TM5 simulated stratospheric $NO_2$ VCDs to the MAX-DOAS measurements in the morning and evening, but kept the shape of the $NO_2$ diurnal cycle in the TM5, then interpolated the  $NO_2$ VCD at the TROPOMI overpass time from the shifted TM5 stratospheric $NO_2$ VCDs.

*Figures 6,10, 11: What quantity are the error bars? It should be mentioned in the caption*

The error bars are the precision of the of TROPOMI tropospheric or stratospheric $NO_2$ VCDs. We have added the explanation in the captions. In Fig.11, the horizontal error bar is the precision of MAX-DOAS $NO_2$ VCD; the vertical error bar is the precision of TROPOMI $NO_2$ VCD.

---

## Author Comment (AC2) · 20 Feb 2020

*The manuscript "Shipborne MAX-DOAS measurements for validation of TROPOMI NO2 products" presents results of NO2 MAX-DOAS measurements from several ship cruises in the Atlantic and Pacific Ocean which are used to validate the NO2 measurements of the new TROPOMI instrument and NO2 simulations of the TM5-MP model. They found a good agreement between the three datasets. In general the manuscriptis well structured and fits in the scope of AMT. Before publication, the following comments should be considered: General comments:*

*Where are the data from the cruise SO201712 shown? Are they only used in Fig.1? If the data are not used for validation, they should not be shown in this manuscript.*

We agree that we did not show the MAX-DOAS data from the cruise SO201712. However we prefer to keep the cruise SO201712 in Fig.1 because it is a summary of all our cruises. It is also interesting to show the dust plume in this trip.

*• discussion of the results of previous studies, for example:*

*– P2, L41: What are the results of the validation measurements over land? Please cite some papers.*

Three references provided by Referee 1 have been cited in the introduction in lines 41-42.

Zhao et al., 2019, Griffin et al., 2019 and Ialongo et al., 2020.

*– P2, L45: What are the results from the previous studies of the comparison between shipborne and satellite measurements?*

The following texts are included in the revised manuscript in lines 48-52.

Peters et al. (2012) found good agreement between morning MAX-DOAS stratospheric $NO_2$ VCDs and the SCIAMACHY and GOME-2A stratospheric $NO_2$ VCDs. Behrens et al. (2019) reported that the GOME-2B stratospheric $NO_2$ VCDs were similar to the morning MAX-DOAS stratospheric $NO_2$ VCDs, while the GOME-2A values were slightly higher than the morning MAX-DOAS stratospheric $NO_2$ VCDs.

*– Sect.4.5: comparison with previous studies is missing*

The following texts are added in sect. 4.5

Similar to Peters et al. (2012) and Behrens et al. (2019) our MAX-DOAS also measured the latitude dependent shape of stratospheric $NO_2$. Because the TROPOMI overpass time is close to noon, we cannot use the morning or evening MAX-DOAS values to compare with TROPOMI data. The morning and evening MAX-DOAS $NO_2$ VCDs were calculated from SZA of 88 to 92 degree by Peters et al. (2012) and Behrens at al.(2019). We can only use the NO2 $VCD_S$ until the solar zenith angle of 89 degree. Peters et al. (2012) reported that the tropospheric $NO_2$ could only reach detect limit when there were ship emission. This agrees with our tropospheric $NO_2$ measurements.

*• The aerosol data are described in Sect.2.1.3. Is there a relationship between aerosol load and differences between the satellite and MAX-DOAS measurements?*

We did not see the relationship between aerosol loading and the difference between TROPOMI and MAX-DOAS stratospheric $NO_2$. Perhaps we could see it if there were high aerosol loading and high tropospheric $NO_2$.

*Specific comments:*

*• throughout the manuscript: slant/vertical column→slant/vertical column density.*

We have changed it in the revised manuscript.

*• throughout the manuscript: "R.V."→"RV"*

We have changed it in the revised manuscript.

*• P3, L67: absolute or relative wind speed and wind direction?*

There are both absolute and relative wind speed and wind direction in the ship data. We revised the texts by including absolute and relative wind speed and wind direction.

*• P3, L73: "The ships were quite stable measurement platform, ….." : Is this really true? For all cruises, pitch and roll angles in a range of±1∘ sounds for me a little bit unrealistic, especially for the cruise SO201712.*

The pitch is mostly with +/- 1 degree. The roll is slightly larger than the pitch, mostly smaller than +/- 2 degree.

*For the cruise SO201712, the pitch was also mostly within +/- 1 degree. There were one or two days that the rolls were larger than +/-2 degree. We have changed the roll to +/- 2 degree in the revised manuscript.*

*• P3, L79-81: Please move the part of the aerosols to Sect. 2.1.3*

We have removed the two lines.

• *P4, L89-97: The whole set-up of the instrument remains unclear to me. Did the instrument really pointed to the backward? Are the 200° clockwise or counterclockwise? Why are measurements to the back of the ship used? I would assume that these measurements are mostly contaminated by the ship plume.*

The 200 degree is clockwise. We have added clockwise in the revised manuscript.

The MAX-DOAS measures both forward and backward directions. The measurements were contaminated by the ship emissions when the wind was from the back of the ship but we could use the measurements from the forward direction. When the wind was from the front of the ship, we did not see the ship emissions.

• *P4, L99: What is about the ship movement within one minute? How large is the error which is introduced in the elevation angle?*

We had some ship data at 1s time interval from the first trip from 17 December 2017 to 9 Jan 2018. The following figures show an example of four minutes (240 s) pitch and roll of the ship Sonne on 4 January 2018. The data were sampled at 1 s interval. The pitch and roll of the ship show a periodic pattern. As shown in the figure, there are several periods of pitch and roll in one minute. This suggests that in one minute the averaged pitch or roll of the ship can be much smaller than the pitch and roll sampled at one minute interval. The integration time of the MAX-DOAS measurement is about one minute. We estimate the error in the elevation angle is +/-1 degree.

[Figure]

[Figure]

• *P4, L102-103: Maybe it would be good to move this sentence to the description of the position of the instrument.*

We have moved the sentence in line 105-106 after describing the measurement angles.

• *P4, L120: Please add also the coordinates in °E.*

We added the longitudes, 20 degree west on 25 Dec 2017 and 23 degree west on 30 Dec 2017.

*• P4, L118: Why is an AOT of 0.05 chosen for days without aerosol measurements?*

The days without aerosol measurements were due to fully cloudy and/or rain. We know the AOT were low from the AOTs in the days close by. We made a simple assumption of the AOT for all days without aerosol measurements.

*• P5, L129: What are detailed results? Please add some further information.*

The sentence is revised as follows.

The TROPOMI $NO_2$ product provides tropospheric, stratospheric, and total vertical column densities (VCDs) and their precision, as well as detailed results for example $NO_2$ slant column densities and precision, airmass factors.

*• P7, L187: What means "later"? Please add an reference.*

We added a reference and changed later to Sect. 4.5. The revised sentence is as follows.

The uncertainty of the AMFs caused by the neglecting of the $NO_2$ photolysis has been shown by (Van Roozendael and Hendrick, 2012) and will be discussed in Sect. 4.5.

*• P8, L237: Please add a tick in Fig.3 at 70∘? Than it is easier to see.*

A tick is added at 70 degree in Figs. 2 and 3.

*• P9, L270-272: Can these results be presented in a small table?*

A table shows the comparison of TM5 and MAX-DOAS stratospheric $NO_2$ VCDs at different SZA ranges is added. The comparison of TM5 and MAX-DOAS stratospheric $NO_2$ VCDs for SZA between 75 and 89 degree is removed from the original Table 2 because it is included in the new table.

*• P9, L273-276: The TROPOMI results are unexpected as the section is about MAX-DOAS and TM5-MP. Change title of the section or put the paragraph somewhere else in the manuscript.*

The title of the section is changed.

*• P10, L288: What is with the first days (28 Jan. - 2 Feb.) of the cruise?*

The first days of the cruise the ship was in the EEZ. We were not allowed to do MAX-DOAS measurements. This is also the reason we do not have measurements close to harbours.

*• Figure2: Please use a normal time axis (such as local time), because day fraction is hard to understand. Please add "strat." to y-label and add the coordinates of mean ship position form the 5 Feb. 2019 in the caption.*

We have changed the day fraction to local time. The figure and caption have been made as suggested.

• *Figure 3: Please use a normal time axis (such as local time). Please add "tropo.." to y-label and add the coordinates of mean ship position form the 24 June 2019 in the caption.*

We have changed the day fraction to local time. The figure and caption have been changed as suggested.

• *Figure 4: Please use a normal time axis (such as local time).*

We have changed the day fraction to local time.

• Figure 5: Please add a unit to the offset in the figure caption.

The unit molec cm$^{-2}$ is added in the caption.

• Figure 6: Please adjust the y-label to the label of the other figures. Please use the day of month for the x-axis.

We have used day of month in x-axis and adjusted the y-label.

• Figure 7: Please add a unit to the offset in the figure caption. TROPOMI is an acronym. Please use capital letters.

We have added the unit molec cm$^{-2}$ and used capital letters for TROPOMI.

• Figure 8: Please add a unit to the offset in the figure caption. Please change y-label (see Fig. 7).

We have added the unit molec cm$^{-2}$ and used capitcal letters for TROPOMI.

• Figure 9: What means "MAX-DOAS TM5 interpolated"? In L 309 it is written "…MAX-DOAS and TROPOMI stratospheric NO2…".

It should be MAX-DOAS, not TM5 interpolated. We have changed the caption.

• Figure 10: Please add "MAX-DOAS" to the legend. Please adjust y-label (Trop. NO2 VCD) and use day of month for x-axis.

Figure 10 is made according to the suggestions.

• Figure 11: TROPOMI is an acronym. Please use capital letters in the y-label.

We have used capital letters, TROPOMI.

• Table 1: The official cruise names are written with a slash in the name such as SO268/1. Please adjust throughout the manuscript.

We have changed the cruise names.